# Replicating the Hadley Cell edge and subtropical jet latitude disconnect in idealized atmospheric models

Molly E. Menzel[1,2], Darryn W. Waugh[2], Zheng Wu[3,4], and Thomas Reichler[5]

[1]NASA Goddard Institute for Space Studies, New York, New York, United States
[2]Department of Earth and Planetary Sciences, The Johns Hopkins University, Baltimore, Maryland, United States
[3]Department of Atmospheric and Environmental Sciences, University at Albany, SUNY, Albany, New York, United States
[4]Institute for Atmospheric and Climate Science, ETH Zurich, Zurich, Switzerland
[5]Department of Atmospheric Sciences, University of Utah, Salt Lake City, Utah, United States

**Correspondence:** Molly E. Menzel (molly.menzel@nasa.gov)

**Abstract.** Recent work has shown that variability of the subtropical jet's (STJ) latitude, $\phi$STJ, is not coupled to that of the Hadley Cell (HC) edge, $\phi$HC, but the robustness of this disconnect has not been examined in detail. Here, we use meteorological reanalysis products, comprehensive climate models, and an idealized atmospheric model to determine the necessary processes for a $\phi$HC and $\phi$STJ disconnect in the Northern Hemisphere's December-January-February season. We find that a decoupling can occur in a dry general circulation model, indicating that large-scale dynamical processes are sufficient to reproduce the metrics' relationship. It is therefore not reliant on explicit variability in the zonal structure, convection, or radiation. Rather, the disconnect requires a sufficiently realistic climatological basic state. Further, we confirm that the robust disconnect between $\phi$STJ and $\phi$HC across the model hierarchy reveals their differing sensitivities to midlatitude eddy momentum fluxes; $\phi$HC is consistently coupled to the latitude of maximum eddy momentum flux but the $\phi$STJ is not.

## 1   Introduction

There is considerable interest in detecting and predicting tropical expansion as a result of increasing greenhouse gases (Seidel et al., 2008; Birner et al., 2014). Early studies examining tropical expansion used various metrics to define the edge of the tropics, including the poleward extent of the Hadley Cell (HC) as well as the subtropical jet's (STJ) location. However, studies presented contradicting conclusions based on their choice of metrics (Seidel et al., 2008; Davis and Rosenlof, 2012; Davis and Birner, 2013; Birner et al., 2014). Subsequent comparisons then exposed a disconnect between upper tropospheric and lower tropospheric metrics (Solomon et al., 2016; Waugh et al., 2018). Davis and Birner (2017) similarly categorize the upper and lower tropospheric metrics as "zonal circulation" and "meridional circulation" metrics, respectively. One specific result revealed there is no interannual correlation between the STJ latitude and HC edge in reanalyses products or coupled model output (Waugh et al., 2018; Menzel et al., 2019) and they have distinct responses to increased $CO_2$ (Davis and Birner, 2017; Menzel et al., 2019).

Historically, large-scale atmospheric circulation in the lower latitudes has been described by axisymmetric theory. In particular, it is dominated by a thermally direct meridional circulation known as the HC (Lorenz, 1967) where the flow is angular

momentum conserving and the circulation's poleward extent is determined by energetic constraints (Held and Hou, 1980; Lindzen and Hou, 1988). Additionally, the STJ is attributed to the HC's poleward advection of angular momentum. As the HC's upper branch circulates poleward, the zonal-mean zonal wind must increase to maintain angular momentum conservation and accommodate the flow's decrease in distance to the earth's axis of rotation. This has led to a persistent assumption that the STJ is co-located and co-varies with the edge of the HC.

Although useful to conceptualize zonal-mean flow, axisymmetric theory is limited; the presence of eddies at higher latitudes resulting from non-axisymmetric processes proves a strong influence on HC dynamics (Schneider, 2006). Rather than invoking energetic constraints, the HC's meridional extent is instead determined by baroclinic instabilities (Held, 2000) and can be described by a critical latitude whereby the angular momentum conserving flow can no longer remain stable (Walker and Schneider, 2006; Korty and Schneider, 2008). In this vein, HC edge variability is directly related to that of static stability and midlatitude eddies (Davis et al., 2016). Indeed, the HC edge's transient response to atmospheric $CO_2$ follows that of the latitude of maximum eddy momentum flux (Chemke and Polvani, 2019), and is strongly correlated with the eddy-driven jet (EDJ) both interannually and in response to changes in greenhouse gas concentrations. (Kang and Polvani, 2011; Solomon et al., 2016; Davis and Birner, 2017; Staten and Reichler, 2014).

The STJ's relationship with both the HC and midlatitude eddies remains less clear. Despite the logical expectation that the STJ latitude co-varies with the HC edge, there is no empirical evidence to support it (Waugh et al., 2018). Both observations and reanalysis products reveal a discernable poleward shift of the HC edge, but such a trend in the STJ is unsubstantiated (Seidel et al., 2008; Birner et al., 2014). Posing the question, "is the subtropical jet shifting poleward?" Maher et al. (2020) confirm that the lack of trend in the STJ cannot be explained by insufficient methods for STJ detection, nor is it obscured by large STJ variability. Regarding natural variability, Menzel et al. (2019) demonstrate that the HC edge is not correlated with the latitude of the STJ and its relationship with the STJ strength is inconsistent. Interannually, an expanded HC is associated with a weaker STJ but in response to increased $CO_2$, the HC edge shifts poleward and the STJ strengthens (Menzel et al., 2019). Further, the HC edge and STJ strength have differing transient responses to forcing. While the HC edge responds within 7-10 years, similar to the latitude of maximum eddy momentum fluxes (Chemke and Polvani, 2019), the STJ's strength takes 40 years to reach its steady state response (Menzel et al., 2019).

Is the disconnect between the STJ and HC edge a robust result and what is their relationship to the midlatitude eddies? In this study, we use idealized atmospheric modelling to address this question. Specifically, we consider the most basic idealized three-dimensional atmospheric model available, a dry general circulation model, with varying basic states. While there are some unrealistic features with these models, numerous previous studies have demonstrated that they can provide insight into the dynamical interaction between the tropical and midlatitude circulation (Eichelberger and Hartmann, 2007; Sun et al., 2013; McGraw and Barnes, 2016). Each model configuration presented uses a thermal relaxation towards an equilibrium temperature, but range between a zonally-symmetric equilibrium temperature set by an analytic function, and one that is varying in all dimensions and derived to reproduce the observed atmosphere. Not only does idealized modelling allow us to isolate the circulation features' sensitivity to midlatitude eddies, it simultaneously reveals the extent to which a simplified atmosphere can represent the STJ. If none of the dry model simulations can reliably produce a STJ, this would indicate that the STJ's

behavior requires processes not included in the model, such as variability in convective processes or sea surface temperatures. Alternatively, if the model can produce a sufficiently realistic STJ and subsequent disconnect from the HC edge, then the mechanisms involved do not require these processes.

Details regarding these idealized model configurations, along with other method choices made in this study, are included in Section 2. We then consider metric relationships evident in coupled model and reanalysis product output in Section 3, and Section 4 presents results from the varying idealized model configurations. Lastly, the implications and limitations of our study are found in Section 5.

## 2 Models and Methods

For all analyses, we present a focused view of the Northern Hemisphere's (NH) December-January-Februrary (DJF) season. Not only does winter feature a dominant HC compared to summer, spring, and fall, it is also when the STJ is well-separated from the EDJ. This allows for unambiguous detection of all prominent features.

### 2.1 Meteorological Reanalysis Products

In this study, we use three reanalysis products provided by the Stratosphere-troposphere Processes And their Role in Climate (SPARC) Reanalysis Intercomparison Project (S-RIP) (Fujiwara et al., 2017) to examine the "observed" atmosphere; the European Centre for Medium-Range Weather Forecast's ERA5 (Hersbach et al., 2020), the second Modern-Era Retrospective analysis for Research and Applications (MERRA-2) (Bosilovich et al., 2016), and the Japanese Meteorological Agency's Japanese 55-year Reanalysis (JRA-55) (Kobayashi et al., 2015). For all fields, we calculate the DJF seasonal average from the zonal-mean monthly output, consider a 42-year time series, 1980-2021, and detrend the metrics before correlation calculation. The eddy terms are calculated from 6-hourly output, which is also available for all included fields. Note, the MERRA-2 output provided by S-RIP has missing values in certain lower-tropospheric levels. Therefore, the MERRA-2 fields with lower levels relevant to metric calculations (i.e. zonal and meridional wind) are taken directly from the National Aeronautics and Space Administration's Global Modelling and Assimilation Office. Lastly, most analysis of the S-RIP output presents the mean across all three reanalysis products.

### 2.2 Coupled Climate Model Output

In addition to the reanalysis products, we also look at output from coupled climate models that participated in the Climate Model Intercomparison Project, Phase 5 (CMIP5) (Taylor et al., 2012). All analysis is done with the first ensemble member (r1i1p1) of the preindustrial control (piControl) experiment, where the radiative agents of atmospheric composition are held at their pre-industrial levels. We take the zonal-mean monthly output from the same 23 climate models used in Menzel et al. (2019) to calculate the DJF seasonal average and present model-mean results. For the eddy calculation, only 4 of those 23 models make available the daily data required for the eddy calculation. Due to this, all CMIP5 analysis pertaining to the eddy fields presents the model-mean across those 4 models.

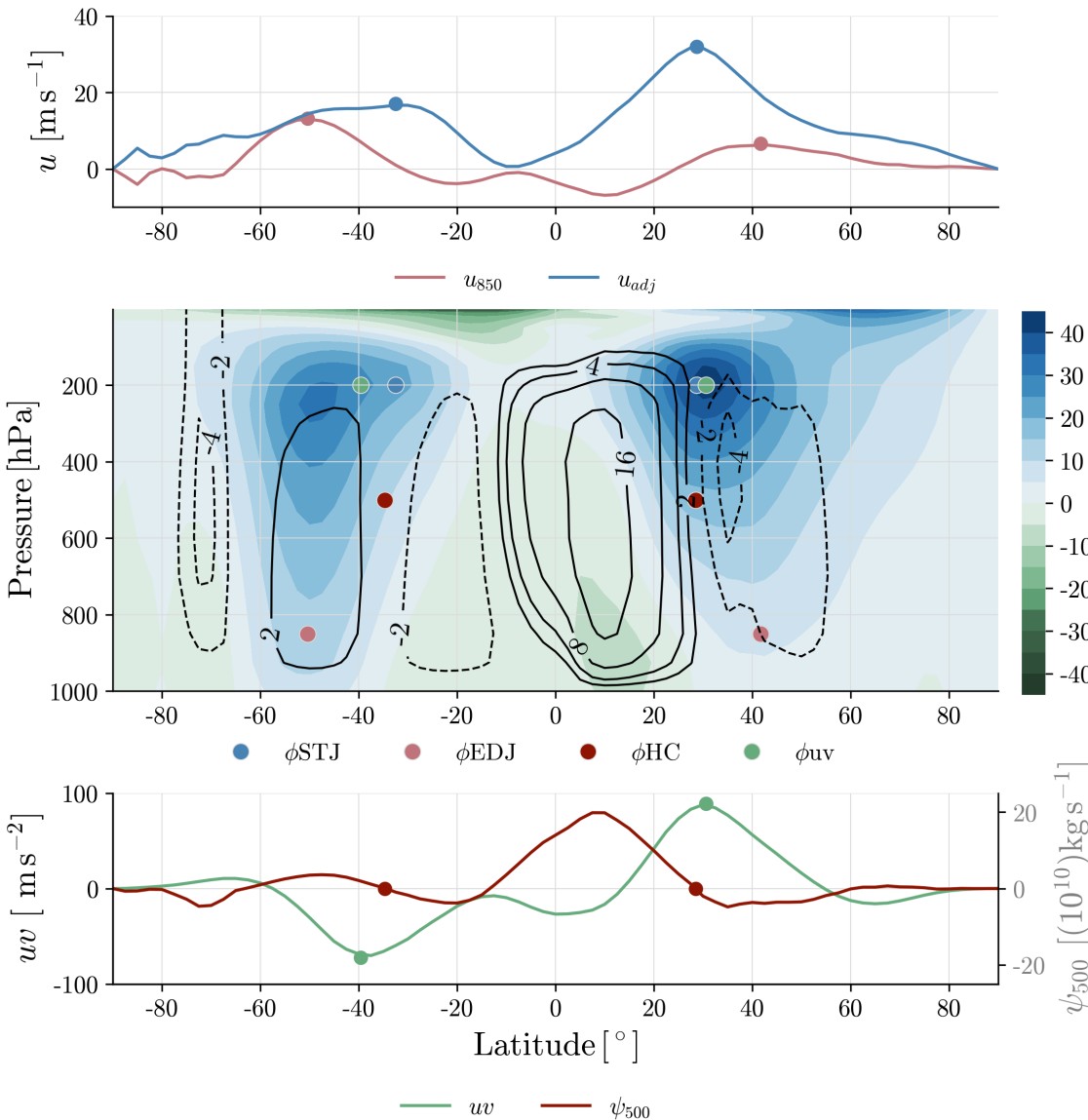

**Figure 1.** DJF zonal-mean climatology of $u_{adj}$ (top, blue), $u_{850}$ (top, pink), the mean meridional streamfunction (middle, black contour lines, $10^{10}\,\mathrm{kg\,s^{-1}}$), the zonal wind (middle, color contours, $\mathrm{m\,s^{-1}}$), $\psi_{500}$ (bottom, red), and $uv$ (bottom, green) for S-RIP from 1979-2019. Each subplot also shows the metric calculated by its corresponding field, $\phi$STJ (top, blue dot), $\phi$EDJ (top, pink dot), $\phi$HC (bottom, red dot), $\phi$uv (bottom, green dot).

## 2.3 Idealized Model Configurations

To diagnose the sensitivity of the HC and STJ to the midlatitude eddies, we perform idealized simulations with a dry atmospheric general circulation model using the Geophysical Fluid Dynamics Laboratory (GFDL) spectral dynamical core in the same configuration as presented in Wu and Reichler (2018). All simulations are forced with a Newtonian relaxation towards one of three different equilibrium temperature profiles.

The most basic simulation replicates that of McGraw and Barnes (2016), hereafter referred to as "MB16." Its equilibrium temperature, $T_{eq}$ is zonally symmetric and set by the analytic function,

$$T_{eq} = max\{T_{strat}, \left[T_0 - \delta_y sin^2\phi + \varepsilon\chi sin\phi - \delta_z log\left(\frac{p}{p_0}\right)cos^2\phi\right]\left(\frac{p}{p_0}\right)^\kappa\} \tag{1}$$

where $T_{strat} = 200\,\mathrm{K}$ is the stratospheric temperature, $T_0 = 315\,\mathrm{K}$, $\delta_y = 60\,\mathrm{K}$ sets the meridional temperature gradient, $\phi$ is the latitude, $\delta_z = 10\,\mathrm{K}$ sets the static stability, $p$ is the pressure, $p_0 = 1000\mathrm{hPa}$ is the reference pressure, and $\kappa = \frac{2}{7}$ is the ratio of gas constant to specific heat of air at constant pressure. This equilibrium temperature deviates from that of Held and Suarez (1994) by its inclusion of $\varepsilon\chi sin\phi$, which simulates a seasonal profile. $\varepsilon$, set to 20K as in McGraw and Barnes (2016), determines the magnitude of hemispheric asymmetry in the temperature profile while $\chi$ modifies that hemispheric asymmetry according to a specific season or month. To simulate the DJF season, we choose $\chi = 0.8796$, the mean of $\chi$ used in McGraw and Barnes (2016) across those months. Note, the configuration still does not simulate a seasonal cycle. Rather, the seasonal conditions are static in time. In later analysis, we modify $\delta_z$ to 15K, 20K, 25K, and 30K, changing the simulated static stability to improve the configuration's basic state. This allows us to test the sensitivity of the circulation features' relationships to this parameter choice.

To improve the basic state of the simulated atmosphere in a dry model, Wu and Reichler (2018) present a new equilibrium temperature field that is derived by iteration to reduce the temperature error, as determined by the MERRA-2 (Bosilovich et al., 2016). Its equilibrium temperature is zonally varying and includes seasonality. Since the equilibrium temperature is developed to simulate observed atmospheric temperature, one may infer that it includes implicit impacts of convective and moist processes. This may be, but the simulation lacks variability of convective and moist processes and only reflects their impacts to setting the basic state. We will refer to this simulation as "WR18."

Here, we introduce an intermediate equilibrium temperature profile that, like WR18, is also derived by iteration but designed to provide a zonally symmetric forcing. The appeal of this setup is that it is closer to the simplicity of MB16 while producing an improved basic state similar to that of WR18. However, simply taking the zonal mean of the WR18 forcing temperature produces a drastically unrealistic atmosphere, with 4 overturning cells in a hemisphere, strong wind jets in the subtropics and polar latitudes, and a corresponding easterly-westerly-easterly-westerly zonal-mean zonal surface wind pattern. Due to this, creation of the zonally symmetric equilibrium temperature file required the same iterative process as that of WR18, reducing the error of the simulated atmosphere according to climatology of MERRA-2. This simulation also allows for seasonality and will be referred to as "WR18z."

All simulations exclude moist and radiative processes, have no topography, and lack any coupling to other climate realms (i.e. ocean, sea ice, land). Note, the equilibrium temperature for WR18 and WR18z were iterated and optimized with topography, but

we have set flat conditions in our simulations. The relaxation time for all idealized configurations is calculated as a function of pressure and latitude. The specific formula used for the MB16 configuration can be found in Held and Suarez (1994). Likewise, refer to Jucker et al. (2014) for the relaxation time used in WR18 and WR18z. Since the dry general circulation model reaches equilibrium quickly, only the first year is excluded in analysis and climatologies are calculated averaging over the remaining 99 years.

## 2.4 Metrics

For metric calculations, we use the TropD python package (Adam et al., 2018) where applicable. Most metrics are calculated using the seasonal- and zonal-mean fields from monthly output. To calculate the eddy terms in the idealized simulations, we use 6 hourly output and then average the eddy field seasonally and zonally. For all metrics locating a maximum of a field, we apply a quadratic fit to the profile as is done in Menzel et al. (2019). Calculation methods for all metrics can be visualized by Figure 1.

The latitude of the EDJ ($\phi$EDJ) is found by using TropD_Metric_EDJ to locate the maximum of the the 850 hPa zonal-mean zonal wind, $u_{850}$ (Fig. 1, top, pink). To locate the STJ, we use the "adjusted" method of TropD_Metric_STJ. This method calculates an adjusted wind field, $u_{adj}$, such that $u_{850}$ is subtracted from the zonal-mean zonal wind vertically averaged between 100-400 hPa (Fig. 1, top, blue). Using the adjusted wind field reduces the signal of the EDJ on the upper tropospheric winds and therefore better distinguishes the STJ from the EDJ. A comprehensive discussion in Adam et al. (2018) states that the adjusted wind method presents a notable difference in the resulting metric and it is more representative of the STJ latitude than by only considering the upper tropospheric wind. Then, rather than simply finding the max of $u_{adj}$, we define the STJ position ($\phi$STJ) as the most equatorward peak of that field. Particularly in the idealized simulations, the adjusted wind may display one weak peak in the subtropics and one strong peak in the midlatitudes. Finding the equatorward peak further mitigates masking by a strong EDJ, enabling proper STJ detection.

We find the HC edge ($\phi$HC) using the "Psi_500" metric in TropD_Metric_PSI. This method defines $\phi$HC as the latitude at which the mean meridional streamfunction at 500 hPa, $\psi_{500}$, crosses zero just north and south of the equator (Fig. 1, bottom, red).

Following the example of Chemke and Polvani (2019), we also find the latitude of maximum eddy momentum flux, $\phi$uv, (Fig. 1, bottom, green) throughout the troposphere, where the eddy momentum flux is defined as $\left[\overline{u^+v^+}\right]cos\phi$ and includes both the transient and stationary eddy terms (i.e. $\left[\overline{u^+v^+}\right] = \left[\overline{u^*v^*}\right] + \left[\overline{u'v'}\right]$ where $[u]$ denotes the zonal mean, $\overline{u}$ denotes the monthly mean, $u^*$ denotes deviations from the zonal mean, and $u'$ denotes deviations from the monthly mean).

In calculating correlations between metrics, years are ignored if one of the metrics is not detectable. This is the case if no peak in the adjusted wind profile is equatorward of $\phi$EDJ. We first calculate the seasonal-mean of metrics for each year to correlate across a time series of that season alone. In the case of the MB16 configurations that simulate the DJF season for all time, we follow this same protocol but average the correlations calculated from a time series of each "season" (e.g. months 1-3, months 4-6, months 7-9, and months 10-12). The resulting variability is comparable to the variability found in the other configurations. Correlations are defined as significant by a p-value test at a 95% confidence interval (i.e. $P \leq 0.05$).

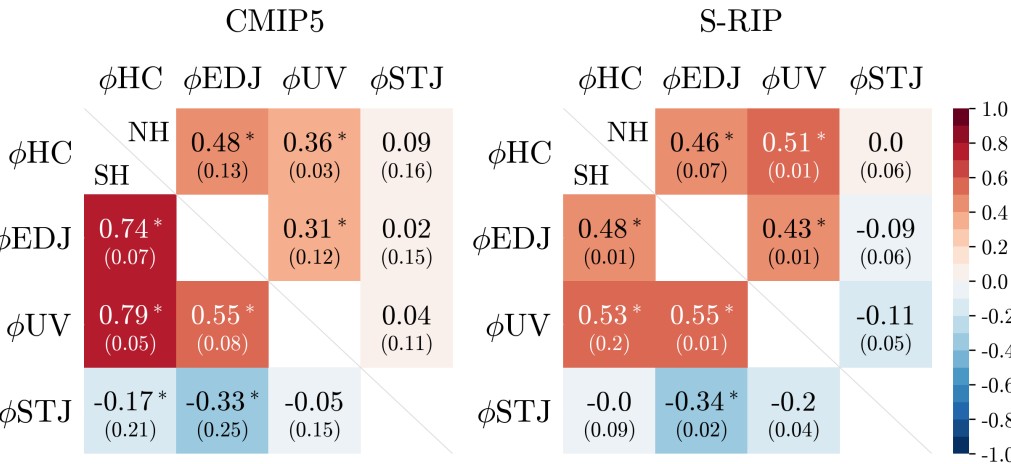

**Figure 2.** SH (bottom left) and NH (top right) interannual correlations for the DJF season of CMIP5 (left) and S-RIP (right). All correlations are the model- or product-mean, the number in parentheses indicates model- or product-spread, and the asterisk denotes that correlations are statistically significant.

## 3 Coupled Models and Reanalyses

Before we analyze the idealized model simulations discussed above, we revisit the interannual HC and STJ relationship in meteorological reanalysis products and coupled climate models. As discussed in the introduction, previous work has shown that $\phi$HC is tied to $\phi$EDJ (Kang and Polvani, 2011; Davis and Birner, 2017; Staten and Reichler, 2014), but the STJ's behavior is distinct from both (Waugh et al., 2018; Menzel et al., 2019). This is illustrated in Figure 2 for the DJF season. Both the reanalysis products and climate models show a near zero correlation between $\phi$STJ and $\phi$HC for both hemispheres, but $\phi$HC has a significant positive correlation with $\phi$EDJ.

We also find low correlations (R < 0.5) between $\phi$STJ and $\phi$EDJ in each hemisphere. Interestingly, there are spurious negative correlations in the SH from frequent masking of the STJ by the EDJ. When $\phi$EDJ is sufficiently equatorward, the two jets become merged, the midlatitude peak in the adjusted wind profile overshadows the peak in the subtropics, and $\phi$STJ is detected at a more poleward latitude due to its proximity to $\phi$EDJ. However, in a more separated state when $\phi$EDJ is sufficiently poleward, the adjusted wind profile has a distinct peak in the subtropics, allowing for easy detection of $\phi$STJ at its more climatological, i.e. equatorward, location. This oscillation between a merged state ($\phi$EDJ is equatorward, $\phi$STJ detected poleward) and a separated state ($\phi$EDJ is poleward, $\phi$STJ climatologically equatorward), gives rise to a negative correlation. Note, the negative correlations are more prominent in SH DJF as the STJ it typically weaker in summer than winter and thus more vulnerable to EDJ behavior. This behavior is also evident when using the default $\phi$STJ metric of TropD as in Menzel et al. (2019), where the $\phi$STJ is defined as the location of maximum $u_{adj}$ rather than the most equatorward peak. In that case, the model-mean negative correlation between $\phi$STJ and $\phi$EDJ is mitigated by more positive correlations of certain models.

Although the lack of coupling between $\phi$HC and $\phi$STJ has been noted, the physical mechanisms responsible for the discon-
nect remain unknown. One compelling suggestion, proposed by Davis and Birner (2017), is that the difference is due to the
meridional streamfunction, used to define the HC edge, being physically linked to the distribution of eddy momentum fluxes.

To see this, first consider the zonal-mean zonal momentum equation expressed by Equation 14.4 in Vallis (2017),

$$\frac{\partial \overline{u}}{\partial t} - \left(f + \overline{\zeta}\right)\overline{v} + \overline{w}\frac{\partial \overline{u}}{\partial z} = -\frac{1}{a\cos^2\phi}\frac{\partial}{\partial \phi}\left(\left[\overline{u^+v^+}\right]\cos^2\phi\right) - \frac{\partial\left[\overline{u^+w^+}\right]}{\partial z} \tag{2}$$

where $\overline{u}$ is the zonal-mean zonal wind, $f$ is the Coriolis parameter, $\overline{\zeta}$ is the zonal-mean relative vorticity, $\overline{v}$ is the zonal-mean
meridional wind, $\overline{w}$ is the zonal-mean vertical wind, $a$ is the radius of the earth, $\phi$ is the latitude, and $\left[\overline{u^+v^+}\right]\cos\phi$ is the eddy
momentum flux.

We may neglect vertical advection and vertical eddy terms such that the equation simplifies to the second term on the left
hand side and the first term on the right hand side. Close to the equator, eddies are considered negligible and the meridional
flow is angular momentum conserving, i.e. the second term on the left hand side equals zero. However, eddy momentum diver-
gence, the first term on the right hand side of Equation 2, becomes increasingly prevalent at higher latitudes. In those regions,
the meridional flow is no longer angular momentum conserving but rather the poleward advection of angular momentum is
balanced by the eddy momentum divergence.

In Figure 2, we see that $\phi$HC positively co-varies with $\phi$uv with significance in both hemispheres. This supports the sugges-
tion that at $\phi$HC, the meridional flow is influenced by eddies (Walker and Schneider, 2006; Korty and Schneider, 2008; Davis
and Birner, 2017; Chemke and Polvani, 2019).

On the other hand, variability of the STJ only relates to HC dynamics where the meridional flow is angular momentum
conserving. Although angular momentum conservation is more prominent in the winter than summer HC, the meridional flow
is never angular momentum conserving at the HC's poleward extent. The result is that while the poleward flank of the HC has
a direct dynamical relationship to the midlatitude eddies via meridional flow, the STJ does not. This could explain why the
correlations between $\phi$STJ and $\phi$uv are less than 0.2.

Clearly, there is a distinction between $\phi$STJ and those metrics associated with meridional flow where the flow is influenced
by eddies (i.e. $\phi$HC, $\phi$uv, $\phi$EDJ). At $\phi$HC, meridional flow is less dependent on angular momentum advection, thus, the
expected coupling between $\phi$HC and $\phi$STJ via angular momentum conservation breaks down.

Further, the disconnect between $\phi$HC and $\phi$STJ and the link between $\phi$HC and midlatitude eddies is found in response to
$CO_2$ forcing. Chemke and Polvani (2019) show that in response to a quadrupling of $CO_2$, the southern hemispheric (SH) shifts
of $\phi$HC and $\phi$uv are correlated (R = 0.68 in the annual mean) and have the same rapid transient response to atmospheric $CO_2$
forcing ($\sim$ 7 years). In response to the same forcing, the STJ shifts poleward minimally and instantaneously while strengthening
with a slower transient response of about 40 years (Menzel et al., 2019).

 **4  Idealized Modelling**

The disconnect between the $\phi$STJ and $\phi$HC shown in Section 3 is a robust result across coupled models and reanalysis products. But, it is not known which physical mechanisms are responsible for the result. To identify which model processes are necessary to replicate the $\phi$STJ and $\phi$HC relationship, we start with the most basic idealized atmospheric model, the dry general circulation model presented in MB16, and increase the model's complexity with WR18 and WR18z. Subsequently, we

 modify the MB16 configuration, improving its simulation of the subtropical circulation.

**4.1  Analytic Equilibrium Temperature**

We first consider the the most basic idealized model, MB16. Comparing its climatological basic state with that of S-RIP, Figure 3 shows that MB16 produces an atmosphere with the relevant circulation features. The temperature decreases with latitude and altitude (Fig. 3, bottom left), there are distinct Hadley and Ferrel Cells, and the zonal winds increase with height (Fig. 3,

 bottom right). However, MB16 differs from the S-RIP climatology in notable ways; the zonal winds are more barotropic and their maximum is located at the top of the Ferrel Cell rather than on the edge of the HC (Fig. 3, bottom right). Additionally, the meridional streamfunction does not extend as high in the atmosphere as that of S-RIP (e.g. the $8\left(10^{10}\right)$ kg s$^{-1}$ contour line is as high as 200 hPa in S-RIP but only reaches 300 hPa in MB16).

What, then, is the resulting relationship between $\phi$STJ and $\phi$HC in MB16 with a default parameter of $\delta_z = 10$? Figure 4

 (dark red) shows that the MB16 produces a positive correlation between $\phi$HC and $\phi$STJ of about 0.66 Also, $\phi$HC and $\phi$STJ both have a significant positive correlations with $\phi$uv, indicating that all features are strongly coupled together and set by the midlatitude eddies. Although such a strong correlation between $\phi$STJ and $\phi$HC is in line with simple angular momentum conservation consideration, it is a strong contrast to the reanalysis product and coupled model output where their correlations are low (see Fig. 4, black and purple). Therefore, such a basic idealized atmospheric model as MB16 is unable to replicate the

 $\phi$STJ and $\phi$HC relationship evident in more realistic climatologies.

**4.2  Derived Equilibrium Temperature**

Above we found that there is a coupling between $\phi$HC and $\phi$STJ in an idealized atmospheric model that uses an analytic equilibrium temperature profile, but does it exist in a model with a more realistic atmosphere? The simulated atmosphere of WR18, where the equilibrium temperature is derived iteratively to replicate that from MERRA-2, is shown in Fig. 3. By design,

 the simulation produces an improved basic state compared to MB16. The zonal wind profile shows larger baroclinicity and the distinct maximum in the upper troposphere is co-located with the HC edge (Fig. 3, middle top right). Additionally, the winter HC strength is relatively stronger than that of the summer HC and winter Ferrell Cell when compared to MB16. However, some features remain inconsistent with S-RIP. For instance, its meridional streamfunction is reduced in strength in the lower latitudes. Additionally, and similar to MB16, the meridional streamfunction does not reach as high in the tropics as in S-RIP.

 Not shown in this climatology, high latitude zonal winds poleward of $60°$S in WR18 have high variability, impacting features in the midlatitudes.

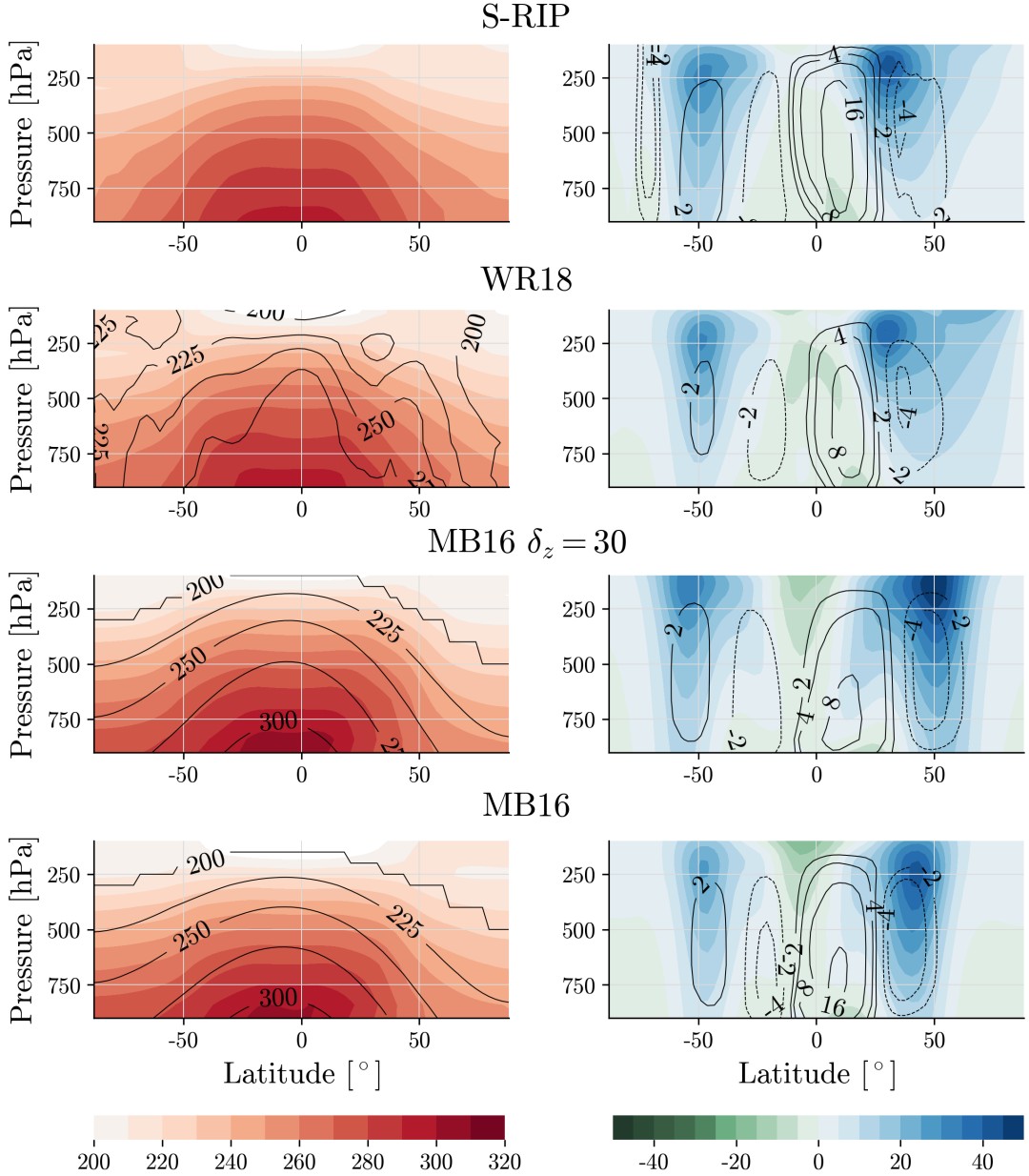

**Figure 3.** DJF zonal-mean equilibrium temperature (left, black contour lines, K) and DJF climatology of the simulated temperature (left, color contours, K), zonal wind (right, color contours, $\mathrm{m\,s^{-1}}$), and mean meridional circulation (right, black contour lines, $10^{10}\,\mathrm{kg\,s^{-1}}$) for S-RIP (top), WR18 (middle top), MB16 ($\delta_z = 30$) (middle bottom), and MB16 (default, $\delta_z = 10$) (bottom).

The improved atmospheric setup in WR18 produces correlations between $\phi$HC and $\phi$STJ that deviate from strongly positive (Fig. 4, blue) as they are less than $0.1$ and insignificant. Meanwhile, $\phi$HC stays significantly positively correlated with $\phi$uv, but

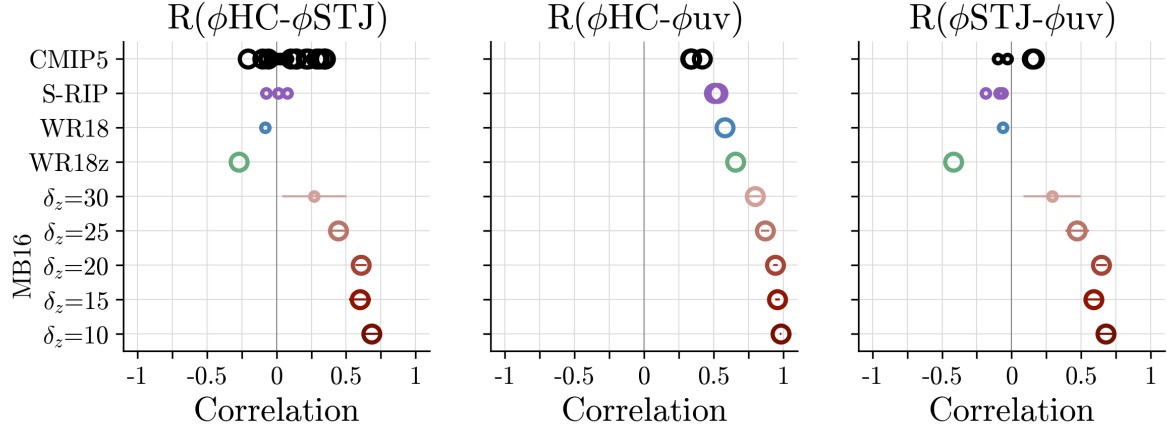

**Figure 4.** NH DJF interannual correlations between the stated metrics for all model configurations. Here, error bars denote one standard deviation across simulated "seasons" (i.e. MB16 which simulates DJF statically). The larger circles denote correlations found to be significant with 95% confidence ($P \leq 0.05$), and the smaller circles denote insignificant correlations.

the correlation between $\phi$STJ and $\phi$uv also reduces to less than $0.1$. This result, that $\phi$STJ and $\phi$HC are not positively correlated
in WR18, reveals that a disconnect between $\phi$STJ and $\phi$HC is possible in a fully dry atmospheric model. A disconnect is therefore not necessarily dependent on variability in more complex processes, such as convection or radiation.

Does it instead depend on zonal asymmetries in the model's forcing? We explore this by considering WR18z, where a new equilibrium temperature field is derived to be zonally symmetric. In the zonal-mean climatology, WR18z produces a similar basic state as WR18 (see Fig. S1 of the Supporting Document). The most apparent differences between the WR18 and WR18z
equilibrium temperature are in the lower troposphere at the SH's high latitudes and the NH's midlatitudes, where WR18z appears more variable. Yet, the mean meridional circulation and zonal wind patterns are close to that of WR18. The only subtle differences are that in WR18z compared to WR18, the magnitude of zonal winds in the upper troposphere is larger, and the meridional streamfunction is weaker in the SH summer but stronger in the NH winter.

The resulting correlations between metrics in WR18z are categorically similar to WR18 (see Fig. 4, green). Although signif-
250 icantly moderately negative, the correlations between $\phi$STJ and $\phi$HC still contrast the strong positive correlations in MB16 and are within the range of correlations from CMIP5. Recall, the moderately negative correlations between $\phi$STJ and $\phi$HC likely reflect occasional masking of the STJ by the EDJ, see Section 3. As in WR18 and MB16, $\phi$HC is positively correlated with $\phi$uv, but $\phi$STJ's correlation with $\phi$uv is significantly moderately negative. So, a $\phi$STJ and $\phi$HC disconnect is not the result of zonal variability in the model's forcing.

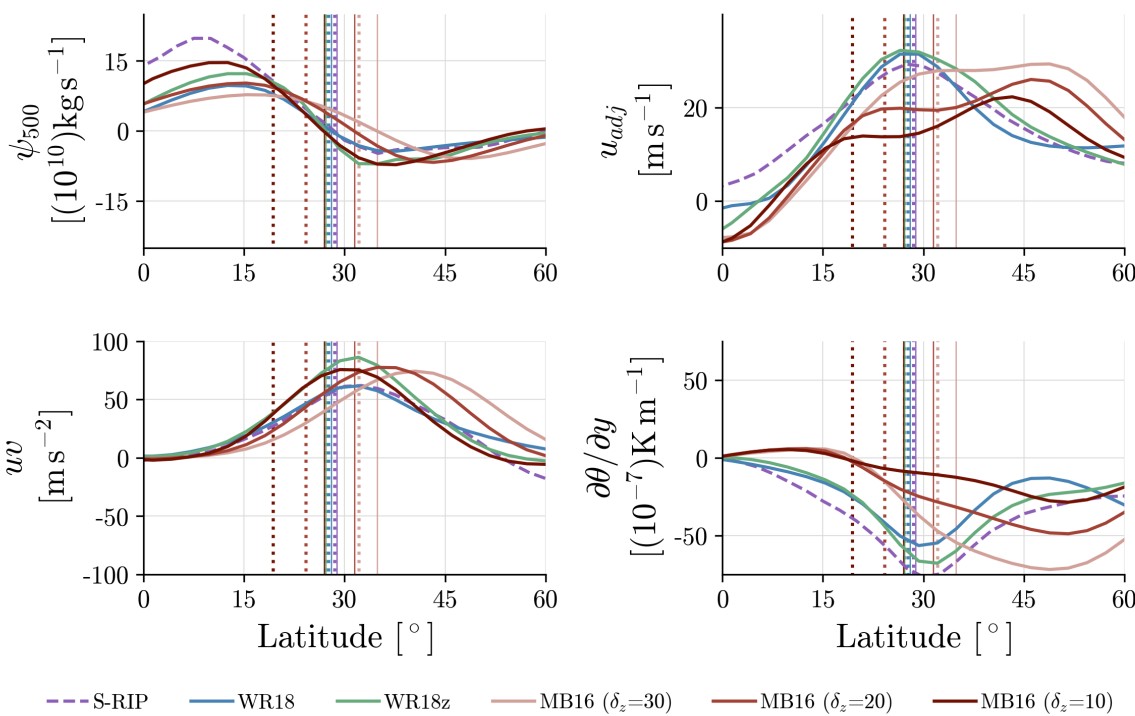

**Figure 5.** NH DJF zonal-mean meridional streamfunction at 500 hPa (top left), adjusted wind (top right), vertically averaged eddy momentum flux between 200-400 hPa (bottom left), and vertically averaged meridional temperature gradient between 100-400 hPa (bottom right) for S-RIP, WR18, WR18z, MB16 (default, $\delta_z = 10$), MB16 ($\delta_z = 20$), and MB16 ($\delta_z = 30$). The dotted and solid thin vertical lines show the climatological $\phi$STJ and climatological $\phi$HC, respectively, for each corresponding simulation.

### 4.3 Modified Analytic Equilibrium Temperature

Given that a decoupling between $\phi$STJ and $\phi$HC is not the result of variability in moist or radiative processes, nor is it the result of zonal variability in the model's forcing, is it possible to replicate the disconnect in a MB16 configuration by improving its basic state?

We explore this by varying $\delta_z$ in Equation 1 from its default value of $10\,\mathrm{K}$ to $15\,\mathrm{K}$, $20\,\mathrm{K}$, $25\,\mathrm{K}$, and $30\,\mathrm{K}$. Physically, increasing this parameter decreases the static stability of the atmosphere, as seen by the lifting of the equilibrium temperature contours in Figure 3 (see the middle bottom left plot). Figure 3 also shows the impact a larger $\delta_z$ has on the basic state. The increase in temperature at lower latitudes relative to higher latitudes increases the meridional temperature gradient. This, via thermal wind balance, increases the zonal winds aloft and gives hints of larger baroclinicity in the subtropics. Interestingly, the tropical meridional circulation is weaker in strength compared to that of MB16.

A more specific visualization of relevant basic state fields across most model configurations can be seen in Figure 5. Perhaps unsurprisingly, the more realistic configurations, WR18 and WR18z, better match the $u_{adj}$ profile seen in S-RIP. There is a

distinct peak in the subtropics, and $u_{adj}$ non-monotonically decreases until reaching about 45°N. In contrast, the three MB16 configurations shown all reveal a larger peak of $u_{adj}$ in the midlatitudes relative to the subtropics. With a larger $\delta_z$ parameter, the strength of $u_{adj}$ in the subtropics increases to a similar magnitude as found in S-RIP, but never to the point of being the dominant peak.

The differences in $u_{adj}$ between idealized models mirrors similar differences in the upper tropospheric meridional temperature gradients, $\partial\theta/\partial y$. Both WR18 and WR18z configurations mimic the S-RIP pattern of $\partial\theta/\partial y$ at lower latitudes, but do not reach the same magnitude. In contrast, all MB16 configurations produce positive $\partial\theta/\partial y$ values until about 20°N. In the subtropics, MB16 ($\delta_z = 30$) is able to produce the strongest $\partial\theta/\partial y$ of all MB16 configurations, closer to both WR18 configurations and S-RIP. However, none of the MB16 configurations are able to produce comparable values of $\partial\theta/\partial y$ to S-RIP at lower latitudes which, by thermal-wind balance, is consistent with their inability to simulate a robust subtropical jet (Fig. 3).

Note, the differences across model configurations are much smaller for the the eddy momentum flux ($uv$) and meridional streamfunction ($\psi_{500}$) fields. This implies all idealized model configurations are adequate in simulating the midlatitude circulation.

Changes to the basic state shown in Fig. 5 are enough to impact the relationship between $\phi$STJ and $\phi$HC (see Fig. 4). As $\delta_z$ increases to $30\,\mathrm{K}$, the significant positive correlation between $\phi$STJ and $\phi$HC reduces to become insignificant and low ($R \sim 0.25$). This is within the range of $\phi$STJ and $\phi$HC correlations found in the CMIP5 models. Similarly, the correlation between $\phi$STJ and $\phi$uv reduces to about $0.25$ and becomes insignificant as well. All the while, $\phi$HC remains positively, significantly correlated with $\phi$uv.

To summarize, the relationship between $\phi$HC and $\phi$STJ as shown by coupled model and reanalysis product output can be replicated in a fully dry atmospheric model without variability in moist or radiative processes, or zonal structure of the forcing. This is supported by the lack of strong positive and significant correlations between $\phi$STJ and $\phi$HC in the WR18, WR18z, and MB16 ($\delta_z = 30$) configurations. The degradation of the significant positive correlations found in the default MB16 configuration occurs as the basic state improves such that a true STJ emerges in the zonal wind profile. Meanwhile, $\phi$HC's strong and significant correlation with $\phi$uv is consistent across the entire model hierarchy and $\phi$STJ's correlations with $\phi$uv mirror those correlations between $\phi$STJ and $\phi$HC for each configuration.

## 5 Concluding Remarks

Altogether, we show that a disconnect between the STJ latitude ($\phi$STJ) and HC edge ($\phi$HC) is robust across a hierarchy of models and does not require simulated variability in convective or radiative processes, or a zonally asymmetric basic state. The simulations that oppose this result present such weak zonal winds in the subtropics that the detected STJ is uncharacteristic of its climatological behavior. This is the case for the MB16 configurations with larger values for tropical static stability. As the basic state improves, in the case of the MB16 configurations with decreased static stability in the tropics, a representative STJ emerges and its disconnect from the HC edge and midlatitude eddies remains consistent with increasing model complexity.

This analysis further reveals that the robust nature of a $\phi$STJ and $\phi$HC disconnect is the result of differing sensitivities to the
midlatitude eddies. For all levels of complexity, $\phi$HC remains significantly and strongly correlated to the latitude of maximum eddy momentum flux ($\phi$uv). The coupling of $\phi$HC and $\phi$uv reflects theory that describes the HC's poleward extent as determined by baroclinic instabilities (Held, 2000; Schneider, 2006; Korty and Schneider, 2008) rather than energetic constraints (Held and Hou, 1980).

In contrast, the STJ is less sensitive to the midlatitude eddies, as evident in the reduced correlations between $\phi$STJ and $\phi$uv
given improved basic states. This is not to say the STJ is entirely unrelated to the midlatitude eddies, rather that their connection is not strong in the zonal-mean, climatological DJF season. Our results leave room for a dynamical relationship between the two features for given regions, or during certain modes of climate variability. An extension of this work to consider those aspects would provide a more detailed view of interaction between the STJ and midlatitude eddies.

Although our paper identifies a disconnect via interannual correlations, correlations alone may not fully encompass the
lack of coupling between $\phi$STJ and $\phi$HC. However, prior studies support the conclusion based on the features' response to $CO_2$ forcing (Solomon et al., 2016; Davis and Birner, 2017; Menzel et al., 2019). One major implication is that the robust lack of coupling between $\phi$STJ and $\phi$HC cautions against conflation of the two metrics. For instance, $\phi$STJ should not be used for detection of tropical expansion if a study's interest is in regional impacts (Waugh et al., 2018). Likewise, $\phi$HC cannot inform behavior of the upper tropospheric subtropical zonal winds that connect to the stratosphere's Brewer-Dobson Circulation
(Shepherd and McLandress, 2011).

At the same time, we do not imply that there is no connection between the STJ and HC. Indeed, the STJ's strengthening in response to $CO_2$ demonstrates the same seasonal, hemispheric, and transient patterns as that of the HC's upper tropospheric upwelling strength and width (Menzel et al., 2023). Rather, the relationship between the STJ and HC is nuanced and level-dependent.

Lastly, our results support use of an idealized dry general circulation model to study large-scale atmospheric dynamics at lower latitudes. So long as care is taken in parameter choices to simulate a sufficient basic state, inclusion of variability in moist and radiative processes may not be necessary. Such methodological choices are dependent on the research question of interest.

*Code and data availability.* The output from all idealized model simulations are publicly available via Zenodo (https://doi.org/10.5281/zenodo.8144564). The version of the GFDL dry dynamical general circulation model used in this study, along with the equilibrium temperature in the WR18
configuration, can be found at https://github.com/ZhengWinnieWu/WR_simpleGCM. All coupled model and reanalysis output is freely available; CMIP5 output can be found through the Earth System Grid Federation at https://esgf-node.llnl.gov, refer to https://s-rip.github.io/ for S-RIP.

*Author contributions.* This study was conceptualized and designed by MEM and DWW. MEM performed the idealized model simulations, conducted the analysis, and created all figures, with input from DWW. ZW provided input files for certain idealized simulations in collabo-
ration with TR. MEM wrote the initial draft with feedback from all co-authors.

*Competing interests.* The authors declare no competing interests.

*Acknowledgements.* Molly E. Menzel and Darryn W. Waugh would like to acknowledge their collaboration with the International Space Science Institute (ISSI) in Bern, Switzerland, through ISSI International Team project 460, Tropical Width Impacts on the Stratosphere (TWIST). Helpful discussions at ISSI TWIST meetings improved the quality of this work. Molly E. Menzel is supported by an appointment to the NASA Postdoctoral Program at the Goddard Institute for Space Studies, administered by Oak Ridge Associated Universities under contract with NASA. She also acknowledges support from the NASA Modeling, Analysis, and Prediction program. Molly E. Menzel and Darryn W. Waugh both acknowledge support by the U.S. National Science Foundation (NSF) award AGS-1902409 and Thomas Reichler acknowledges support by NSF under award AGS-2103013.

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
