# Peer review of "Supporting Document for “Replicating the Hadley Cell and subtropical jet disconnect in idealized atmospheric models”"

_EGUsphere, 2023_

## Referee Comment (RC1)

Review of

**Replicating the Hadley Cell and subtropical jet disconnect in idealized atmospheric models**

Molly Menzel et al.

**General**

The authors employ a model hierarchy approach to examine the disconnect between the position of the subtropical jet (STJ) and the edge of the Hadley cell (HC), which was found in previous work, and is in contrast to theoretical expectation. The main finding is that the disconnect occurs in a simple dry atmosphere, and therefore does not necessitate processes related to moist convection, radiation, and to some degree, zonal asymmetries in the thermal m background state. The topic is important and timely, and the methodology used is well suited for the research question at hand. However, I find that some potential criticisms of the methodology and findings are not sufficiently addressed. I therefore recommend accepting the paper after a major revision. My comments are listed below.

**Major comments**

1. The argument in lines 165-168 does not make much sense to me. In the upper level, the edge of the HC is found where $f\bar{v}$ vanishes. Therefore, the eddy momentum fluxes term must be balanced by the vorticity flux, rather than "dominating" the balance. This is the gist of the Korty and Schneider (2008, "Extent of Hadley circulations in dry atmospheres" ) argument, which is unfortunately not mentioned in the text. Based on the Korty and Schneider argument, I would like to offer an alternative framing of the problem.

The upper-level zonal momentum balance (Eq. 2) can be rewritten as

$$f\bar{v} - \frac{\bar{v}}{a\cos\phi}\frac{\partial(u\cos\phi)}{\partial\phi} = \frac{1}{\cos^2\phi}\frac{\partial}{\partial\phi}\left(\left[\overline{u^+v^+}\right]\cos^2\phi\right)$$

Under this balance, the EDJ, STJ and HC edge should all be located, because at the latitude where $f\bar{v} = 0$ (HC edge) there exists a solution where the mean momentum flux and eddy momentum fluxes are maximal, so that their gradient vanishes there. Also, this indicates that the peak of $u\cos\phi$ should be used to calculate the positions of the STJ and EDJ. The question then becomes why is this balance not manifest in the metrics? One simple explanation is that since the STJ and EDJ metrics, as well as the $\psi_{500}$ metric, are calculated at different levels, the lack of covariance of the STJ and EDJ metrics is in large part due to their calculation at different levels. Similarly, at the 500 hPa level, mean momentum fluxes are much weaker, and so the dominant balance is between the eddy momentum fluxes and $f\bar{v}$, explaining their closer relation. In conclusion, it is important to reject the null hypothesis that the disconnect between the STJ and HC is not a mere feature of the definition of the metrics (i.,e., calculation at different levels), rather than representing a fundamental physical property of the MOC. The authors conclude that the relationship between the STJ and HC is nuanced and

level-dependent. They should therefore convince the reader that there is more to the results than this level sensitivity of the metrics.

2. Since there is no seasonal signal in the idealized model, the concept of inter-annual variability seems artificial and potentially misleading. The variance would be strongly controlled by the relaxation constant, which is not stated, but is likely small since the authors state that 1 year is sufficient for spinup. Please justify this methodology and the physical meaning of the correlations shown in Figure 4.

3. The 200 hPa isobar in Figure 3 suggests that there are instabilities in the mean state of the MB16 simulations. This likely affects the signal to noise ratio of the results. Please comment on these or consider averaging over a shorter simulation period.

**Minor comments**

1. Shouldn't the title read … Hadley cell edge and the subtropical jet position disconnect …?

2. In Figure 5, it is hard to distinguish between the MB $\delta_z$ 20 and 10 simulations. Also, add in the caption, …thin vertical lines…

**Comments by line**

15

34  $CO_2$ —> changes in greenhouse gas concentrations.

94 and elsewhere     What are you referring to here in terms of "accuracy"? For an idealized model, how can accuracy be defined? I can understand aiming for a mean state that mimics observed conditions. But accurate does not seem like the proper terminology to use here.

111  The rate of relaxation would be determined by the relaxation constant, which is not specified.

138  focused view of the

172  I think you what you mean here is that the relation is stronger in the winter hemisphere, in which the AMC limit is more prominent. This should be clarified.

194  The zonal winds are **more** barotropic compared to the other circulations but are definitely not barotropic.

---

## Referee Comment (RC2)

Replicating the Hadley Cell and subtropical jet disconnect in idealized atmospheric models – reviewing
Molly Menzel, Darryn Waugh, Zheng Wu, Thomas Reichler

General comments:

Menzel et al., use correlations in reanalysis and a range of models to understand the disconnection between the Hadley Cell edge and the subtropical jet latitude. They argue that the disconnection is due to the STJ latitude being closely related to angular momentum conservation, whereas the HC edge is more closely linked to mid-latitude eddies. This is a really interesting study and helps address a gap in our dynamical understanding of the global circulation.

However, I would like to see more analysis or back ground literature to support your findings. Correlations are not sufficient on their own to explain the disconnect, and the correlations are moderate (around r=0.5 in the reanalysis), so could only ever be one part of the story. You do attempt to understand the mechanisms explaining the disconnect by improving the basic model by decreasing its static stability, and find that moist or radiative processes are not relevant. As you do not extend the analysis beyond this point, I was left with the impression that static stability should explain the mechanisms behind the disconnect, without an understanding of why. The profiles of static stability (Fig. 5) in the more complex WR18 and WR18Z simulations are further from that of the reanalysis in the subtropics (around 30 degrees) than the most improved (deltaZ=30) simulation. As there is no correspondingly better correlation between the different metrics (Fig 4), I am also not convinced that static stability could be the whole answer.

I admit I am not familiar enough with Hadley Cell dynamics to determine how much analysis is required, or if citing relevant literature is sufficient, or a combination of both. It may be sufficient to explain the role of static stability more fully and clearly. Alternatively, your argument could be supported by comparing rates of change with the HC edge and midlatitude eddies, or deeper analysis into the different experiments. In either case, I think this paper will be very useful once this additional information is added.

Specific comments:

Line 16: It is unclear what 'them' refers to. I think you mean the upper tropospheric and lower tropospheric metrics? I suggest you reword to make this clearer. Also, not clear if these metrics are for the STJ, HC, or both.

Line 40: Following from previous comment, this hypothesis motivates your study but does not clearly emerge as important from your introduction. I suggest reworking and trimming the introduction to really highlight and support why you are addressing this hypothesis.

Line 42: 'the most idealized ... model' reads strangely. It also doesn't tell me specifically what the model is. In line 186, the model is described as the 'most basic idealized model', is 'basic' missing in line 42?

Line 50/51. Not clear what 'its behaviour' refers to. 'Its' could be the model simulations or the STJ. I think you mean the STJ, but I suggest you reword to make this sentence clearer.

Line 60: You say you use three reanalyses, but only present the results for one. I could not find information around S-RIP that suggests averaging over these three reanalyses. Is that what you have done here? If so, you need to state this. Do the results vary across the three reanalyses?

Line 64: Which season? Later in the text you mention DJF, but it would be good to state here, as well as in the introduction and abstract.

Line 64: Is this data detrended before analysis? A strong trend in the Hadley Cell edge would correlate well with a corresponding trend in the mid-latitude eddies, and may provide misleading results about how well connected the HC edge and eddies are on an inter-annual time scale.

Line 71: Is it appropriate to use CMIP models. Are they suited for looking at large-scale circulation relationships? Why CMIP5 not CMIP6?

Line 84/Equation 1: What do the deltas $\delta$ symbolise? Delta is explained later in the text, but should be covered here.

Lines 138-140 As mentioned in earlier comment, I suggest you move this paragraph describing season used in this study to be in the methods and abstract.

Line 149, and elsewhere where relevant: There are strong seasonal differences in the HC, STJ, EDJ locations and strengths, how do these differences impact your results for the southern hemisphere vs the northern hemisphere?

Line 151 and 153: its not it's

Line 185: Are you planning on finding the physical mechanisms responsible for the disconnection? I do not think you come back to this point. I think you can exclude a couple of mechanisms (moist and radiative processes), but what might explain the physical mechanisms?

Line 195: 'does not' not 'down not'

Line 236: I think it would be good to move (or repeat) the physical interpretation of delta-z in the methods.

Lines 227 – 229 While the correlations do contrast with MB16, they are also quite different to the reanalysis, and I am not convinced being within the range of the CMIP models makes the correlation accurate. Do the statistically significant, if weak, negative correlations between the STJ and HC, and STJ and uv suggest the STJ is more eddy driven in this model? What are the implications for this moderate, negative correlation?

Line 261: This is the first time that deltaZ = 30 has been described as having an improved basic state, and you may wish to state this earlier in the text to make it clearer. How realistic is this deltaZ value? Are there implications for having a much stronger zonal wind in the higher latitudes than the other deltaZ values? The static stability is still much stronger in the tropics than in reanlysis of WR18 or WR18z; is this an issue?

Figure 2: The correlations are the model-mean for CMIP5, are they the mean across the 3 reanalyses produces in S-RIP? Please update the caption accordingly. Do you get very different correlations if you look at the individual reanalyses, or individual models?

Figure 3: It is hard to pick the temperature contours from the colour bar, making it difficult to visually compare to the model equilibrium temperatures. Could a more distinct colour bar be used (e.g. with more colours than shades of red)?

Figure 4. The correlation between Hadley Cell and uv latitudes go off the edge, I suggest you widen slightly. Do you really expect a 100% correlation at deltaZ = 10?.

I wonder if it's helpful to reverse the order of the metrics such that CMIP5 (or preferably reanalysis) is on top and deltaZ = 10 is on the bottom. I intuitively assumed the highest deltaZ was on the top and initially thought increasing static stability made the correlations stronger, the opposite to the real result but an easy mistake to make with a quick glance at the plot. Reversing the order has the added benefit of making the reference value (reanalysis) easier to pick.

Figure 5: Do you find differences in profiles for the southern hemisphere? It might be nice to add to the supplementary material as you show the southern hemisphere data in the earlier plots. To avoid confusion, I also suggest you add 'vertical' to the dotted and solid line description to clarify you are talking about the STJ and HC latitudes, not the stability or wind profiles. What does the CMIP5 profile look like?

---

## Author Response (AR1)

**Response to Reviewers**

Authors' responses to reviewers' comments are distinguished by green text.

**Reviewer 1**

General

The authors employ a model hierarchy approach to examine the disconnect between the position of the subtropical jet (STJ) and the edge of the Hadley cell (HC), which was found in previous work, and is in contrast to theoretical expectation. The main finding is that the disconnect occurs in a simple dry atmosphere, and therefore does not necessitate processes related to moist convection, radiation, and to some degree, zonal asymmetries in the thermal m background state. The topic is important and timely, and the methodology used is well suited for the research question at hand. However, I find that some potential criticisms of the methodology and findings are not sufficiently addressed. I therefore recommend accepting the paper after a major revision. My comments are listed below.

Major comments

1. The argument in lines 165-168 does not make much sense to me. In the upper level, the edge of the HC is found where $f\bar{v}$ vanishes. Therefore, the eddy momentum fluxes term must be balanced by the vorticity flux, rather than "dominating" the balance. This is the gist of the Korty and Schneider (2008, "Extent of Hadley circulations in dry atmospheres") argument, which is unfortunately not mentioned in the text. Based on the Korty and Schneider argument, I would like to offer an alternative framing of the problem.

The upper-level zonal momentum balance (Eq. 2) can be rewritten as

$$f\bar{v} - \frac{\bar{v}}{a cos\phi}\frac{\partial\left(ucos\phi\right)}{\partial\phi} = \frac{1}{cos^2\phi}\frac{\partial}{\partial\phi}\left(\left[\overline{u^+v^+}\right]cos^2\phi\right)$$

Under this balance, the EDJ, STJ and HC edge should all be located, because at the latitude where $f\bar{v} = 0$ (HC edge) there exists a solution where the mean momentum flux and eddy momentum fluxes are maximal, so that their gradient vanishes there. Also, this indicates that the peak of $ucos\phi$ should be used to calculate the positions of the STJ and EDJ.

Thank you for referring us to Korty and Schneider (2008). Indeed, it is a relevant source and we have now cited it throughout the paper (see lines 32, 190, 302). Our results are consistent with the physical relationships discussed in Korty and Schneider (2008): the edge of the Hadley Cell (HC) is strongly sensitive to the eddy momentum fluxes, hence it has high and significant correlations with both the latitude of maximum eddy momentum fluxes and the eddy-driven jet (EDJ) (Figs. 2 and 4). We agree that the edge of the HC (where $\bar{v} = 0$) is found where the eddy momentum fluxes maximize. This mathematical condition dynamically ties the HC edge to the eddy momentum fluxes. Since the EDJ is also dynamically tied to the eddy momentum fluxes and occurs where the convergence maximizes, the HC edge and

EDJ may not be co-located (i.e. they are found at separate latitudes), but they do co-vary interannually. We've modified the language in lines 175-199 to more clearly present these dynamics. Additionally, we have replaced our use of the term "eddy-dominated" with with more appropriate terminology.

With that said, we do not see how this perspective of the zonal-mean momentum balance necessitates that the STJ must therefore co-vary with both the HC edge and EDJ. The condition of $\overline{v} = 0$ and $\frac{1}{cos^2\phi}\frac{\partial}{\partial\phi}\left(\left[\overline{u^+v^+}\right]cos^2\phi\right) = 0$ doesn't inform the structure of $\overline{u}cos\phi$. The STJ's dynamical relationship with the HC occurs in the axisymmetric case where influence of the midlatitude eddies is considered negligible and $f\overline{v} - \frac{\overline{v}}{acos\phi}\frac{\partial(ucos\phi)}{\partial\phi} = 0$. However, as agreed upon above, the midlatitude eddies are relevant at the HC's poleward extent; the meridional flow can no longer be considered angular momentum conserving where $\overline{v} = 0$. Therefore, the necessary condition to couple the STJ to the HC no longer applies at the HC edge. It may be that the HC edge and STJ are located at similar latitudes, but there's no requirement for them to co-vary interannually.

The question then becomes why is this balance not manifest in the metrics? One simple explanation is that since the STJ and EDJ metrics, as well as the $\psi500$ metric, are calculated at different levels, the lack of covariance of the STJ and EDJ metrics is in large part due to their calculation at different levels. Similarly, at the 500 hPa level, mean momentum fluxes are much weaker, and so the dominant balance is between the eddy momentum fluxes and $f\overline{v}$, explaining their closer relation. In conclusion, it is important to reject the null hypothesis that the disconnect between the STJ and HC is not a mere feature of the definition of the metrics (i.e. calculation at different levels), rather than representing a fundamental physical property of the MOC. The authors conclude that the relationship between the STJ and HC is nuanced and level-dependent. They should therefore convince the reader that there is more to the results than this level sensitivity of the metrics.

Thanks for this perspective! Absolutely, it is important to confirm the results are not solely reliant on metric definition. In Figure R1, you'll see a replication of the left subplot in Fig. 4 with $\phi$HC being defined at 200 hPa rather than 500 hPa. When choosing an upper tropospheric level to define $\phi$HC that is comparable to $\phi$STJ's level, the correlations do not improve. Interestingly, the correlations in the MB16 configurations are even lower than those presented in the study.

Additionally, the latitude of maximum eddy momentum flux ($\phi$uv) is a metric that is independent of level choice. It is found at whatever level the maximum eddy momentum flux occurs, typically at 200 hPa. If our results were due to different levels choices, we would expect $\phi$uv to be strongly correlated with $\phi$STJ, and minimally correlated with $\phi$HC. Rather, we find the opposite to be true.

For these reasons, we find the disconnect between the HC edge and STJ latitude to be a physical feature of atmospheric circulation. On this topic, Maher et al. (2020) presents an comprehensive, logical, and convincing argument. As surprising as this result may be based on the Held and Hou (1980) theory, there are numerous studies that support this conclusion

(Seidel et al., 2008; Solomon et al., 2016; Waugh et al., 2018; Menzel et al., 2019). For the purposes of our manuscript, we do not believe it is necessary to replicate those prior studies. Rather, our aim is to reveal what level of model complexity is required to achieve such a disconnect.

Even so, we find that the lack of co-variability between the HC edge and STJ latitude may not be well appreciated within the community. Therefore, we have elaborated on these prior results in the Introduction (lines 37-47).

[Figure]

Figure R1: NH DJF interannual correlations between $\phi$STJ and $\phi$HC where $\phi$HC is calculated at 200 hPa for all model configurations. Here, error bars denote one standard deviation across simulated "seasons" (i.e. MB16 which simulates DJF statically). The larger circles denote correlations found to be significant with 95% confidence ($P \leq 0.05$), and the smaller circles denote insignificant correlations.

2. Since there is no seasonal signal in the idealized model, the concept of inter-annual variability seems artificial and potentially misleading. The variance would be strongly controlled by the relaxation constant, which is not stated, but is likely small since the authors state that 1 year is sufficient for spin-up. Please justify this methodology and the physical meaning of the correlations shown in Figure 4.

Indeed, the seasonality in the MB16 configuration is somewhat artificial. However, the model does sufficiently simulate a December-January-February (DJF) seasonal basic state for all time (i.e. day, months, years). To calculate the correlations for all other configurations with seasonal variability, we first calculate the seasonal mean of the metrics and then correlate across a time series of only one season, in our case DJF. We follow this same protocol for the MB16 configurations but since all months simulate the DJF season, we average the correlations after calculating the correlation across a time series of each "season" (e.g. months 1-3, months 4-6, months 7-9, and months 10-12). The resulting variability is comparable to the variability found in the other configurations and we have added a detailed description of this method at line 152. To confirm that is the case, the variance for all configurations, including the coupled model and reanalysis product output, is shown in Figure R2. For

[Figure]

Figure R2: NH DJF interannual variance of metrics all model configurations.

$\phi$STJ, the WR18 configurations have a comparable variance to the CMIP5 and S-RIP output, between 0.5-1, and the variance in the MB16 configurations are larger, greater than 1. The variance of $\phi$HC is between 0.25-1.25 for all model configurations.

3. The 200 hPa isobar in Figure 3 suggests that there are instabilities in the mean state of the MB16 simulations. This likely affects the signal to noise ratio of the results. Please comment on these or consider averaging over a shorter simulation period.

There are no instabilities in the mean state. The simulated mean-state temperature is shown in the red contour colors and is smoothly varying. The black contour lines show the equilibrium temperature profile that is forcing each idealized model. This includes the 200 K isobar for MB16 and MB16 ($\delta_z = 30$).

Minor comments

1. Shouldn't the title read ... Hadley cell edge and the subtropical jet position disconnect ...?

Yes, thank you for noting that oversight. The revised title is corrected.

2. In Figure 5, it is hard to distinguish between the MB $\delta_z$ 20 and 10 simulations. Also, add in the caption, ...thin vertical lines...

Agreed, we have darkened the MB16 line so that it is more distinguishable from MB16 ($\delta_z$=20). See also Fig. 5's edited caption.

Comments by line

15

See this edit at line 15.

34 CO2 → changes in greenhouse gas concentrations.

See this edit at line 35.

94 and elsewhere What are you referring to here in terms of "accuracy"? For an idealized model, how can accuracy be defined? I can understand aiming for a mean state that mimics observed conditions. But accurate does not seem like the proper terminology to use here.

Thanks for pointing out improper use of "accurate" and "accuracy." We have replaced all occurances of either word with more appropriate terms i.e. "improved basic state" or "sufficient subtropical circulation."

111 The rate of relaxation would be determined by the relaxation constant, which is not specified.

The relaxation constant is a function of latitude and level, we have referred the reader to the relevant literature if of interest, see line 123.

138 focused view of the

See this edit at line 66.

172 I think you what you mean here is that the relation is stronger in the winter hemisphere, in which the AMC limit is more prominent. This should be clarified.

Thanks for your suggestion. That was not our intent of the sentence but we have included that statement in line 193.

194 The zonal winds are more barotropic compared to the other circulations but are definitely not barotropic.

Indeed, thanks for making this point, see our edit at line 215.

**Reviewer 2**

General comments:

Menzel et al., use correlations in reanalysis and a range of models to understand the disconnection between the Hadley Cell edge and the subtropical jet latitude. They argue that the disconnection is due to the STJ latitude being closely related to angular momentum conservation, whereas the HC edge is more closely linked to mid-latitude eddies. This is a really interesting study and helps address a gap in our dynamical understanding of the global circulation.

However, I would like to see more analysis or back ground literature to support your findings. Correlations are not sufficient on their own to explain the disconnect, and the correlations are moderate (around $r = 0.5$ in the reanalysis), so could only ever be one part of the story. You do attempt to understand the mechanisms explaining the disconnect by improving the basic model by decreasing its static stability, and find that moist or radiative processes are not relevant. As you do not extend the analysis beyond this point, I was left with the impression that static stability should explain the mechanisms behind the disconnect, without an understanding of why. The profiles of static stability (Fig. 5) in the more complex WR18

[Figure]

Figure R3: Meridional streamfunction at 500 hPa (top left), adjusted wind (top right), vertically averaged eddy momentum flux between 200-400 hPa (bottom left), and vertically averaged meridional temperature gradient between 100-400 hPa (bottom right) for S-RIP, WR18, WR18z, MB16 (default, $\delta_z = 10$), MB16 ($\delta_z = 20$), and MB16 ($\delta_z = 30$). The dotted and solid thin vertical lines show the climatological $\phi$STJ and climatological $\phi$HC, respectively, for each corresponding simulation.

and WR18Z simulations are further from that of the reanalysis in the subtropics (around 30 degrees) than the most improved ($\delta_z=30$) simulation. As there is no correspondingly better correlation between the different metrics (Fig 4), I am also not convinced that static stability could be the whole answer.

Thanks for sharing these comments. We agree that there is more to the story than just the static stability; the first draft overemphasized its role in explaining the $\phi$STJ and $\phi$HC disconnect. According to prior studies, static stability is one of many proposed mechanisms to explain the behavior of midlatitude circulation in its response to forcing, see Shaw (2019) for a thorough review. Given the focus of the paper is replicating a $\phi$STJ and $\phi$HC disconnect, explaining the nuanced and complex mechanisms controlling $\phi$HC is beyond its scope. Instead, we discern what aspects of the basic state are required to produce a representative STJ such that it's latitude is decoupled from $\phi$HC. In this vein, we have replaced Fig. 5 with one that presents a more comprehensive view of the basic states in all idealized configurations, also included as Fig. R3.

Most notably, and perhaps unsurprisingly, the adjusted wind ($u_{adj}$) profile closely follows the meridional temperature gradients ($\partial\theta/\partial y$). Unlike the WR18 configurations, all MB16 configurations yield positive $\partial\theta/\partial y$ values in the tropics, until about 20°N. MB16 ($\delta_z = 30$)

is able to produce the strongest $\partial\theta/\partial y$ in the subtropics, closer to what is shown in both WR18 configurations and S-RIP. However, as long as a configuration is unable to produce relatively realistic $\partial\theta/\partial y$ at lower latitudes, via thermal-wind balance, it will not be able to sufficiently simulate a STJ.

Also note, the eddy momentum fluxes ($uv$) are not meaningfully different across the shown configurations. This implies that all configurations are adequate in simulating the midlatitude circulation. But, improved simulation of tropical and subtropical circulation is required for STJ behavior and it's subsequent relationship with $\phi$HC.

We have included new discussion of this figure between lines 265 and 279.

Lastly, we acknowledge that interannual correlations alone may not fully encompass a disconnect between $\phi$HC and $\phi$STJ. We do include statistical testing of our correlations in Figs. 2 and 4. Additionally, our extended introduction details prior studies discussing a $\phi$HC and STJ disconnect not just interannually but also in features' response to $CO_2$ forcing (see lines 37-47). We mention this caveat along with said prior studies in the Concluding Remarks section, see line 309. Going beyond these aspects of analysis and literature review is outside the scope of the paper.

I admit I am not familiar enough with Hadley Cell dynamics to determine how much analysis is required, or if citing relevant literature is sufficient, or a combination of both. It may be sufficient to explain the role of static stability more fully and clearly. Alternatively, your argument could be supported by comparing rates of change with the HC edge and midlatitude eddies, or deeper analysis into the different experiments. In either case, I think this paper will be very useful once this additional information is added.

We are optimistic that including Fig. R3 as Fig. 5 presents sufficient additional analysis to highlight aspects of the basic state that are relevant to subtropical circulation.

Specific comments:

Line 16: It is unclear what 'them' refers to. I think you mean the upper tropospheric and lower tropospheric metrics? I suggest you reword to make this clearer. Also, not clear if these metrics are for the STJ, HC, or both.

We have reworded this sentence, see line 16. The references listed consider metrics of circulation features beyond the STJ and HC, but we mention specifically the disconnect between $\phi$HC and $\phi$STJ in the following sentence.

Line 40: Following from previous comment, this hypothesis motivates your study but does not clearly emerge as important from your introduction. I suggest reworking and trimming the introduction to really highlight and support why you are addressing this hypothesis.

Thanks for this suggestion. Based on comments in both reviews, we found it important to elaborate on previous studies that discuss $\phi$STJ and $\phi$HC disconnect (lines 37-47). With that said, since there are now more details relating to the HC, STJ, and midlatitude eddy

relations, we believe the revised paragraph does a better job at setting up the hypothesis in line 48.

Line 42: 'the most idealized ... model' reads strangely. It also doesn't tell me specifically what the model is. In line 186, the model is described as the 'most basic idealized model', is 'basic' missing in line 42?

Yes, thanks for noting this inconsistency. See the edit at line 49.

Line 50/51. Not clear what 'its behaviour' refers to. 'Its' could be the model simulations or the STJ. I think you mean the STJ, but I suggest you reword to make this sentence clearer.

Yes, we were referring to the STJ and have edited the sentence to clarify, see line 57. Thanks for noting the ambiguity.

Line 60: You say you use three reanalyses, but only present the results for one. I could not find information around S-RIP that suggests averaging over these three reanalyses. Is that what you have done here? If so, you need to state this. Do the results vary across the three reanalyses?

S-RIP is an intercomparison of reanalysis products, akin to CMIP5 for coupled models. Therefore, the S-RIP analysis presented is the mean across all three reanalysis products. We have explicitly stated this in line 79. Although we did not test results with individual reanalyses products, differences between them are generally small (e.g. the standard deviation of correlations), and so we believe that variations of results are also small.

Line 64: Which season? Later in the text you mention DJF, but it would be good to state here, as well as in the introduction and abstract.

We have edited the text to clarify consideration of the DJF season, see 74. We've also mentioned our focus on the DJF season in in line 66.

Line 64: Is this data detrended before analysis? A strong trend in the Hadley Cell edge would correlate well with a corresponding trend in the mid-latitude eddies, and may provide misleading results about how well connected the HC edge and eddies are on an inter-annual time scale.

In the original manuscript, the reanalysis product output had not been detrended before calculating the correlations. However, we have since detrended the output as to remove any possible trends in the circulation metrics (e.g. $\phi$HC). This is now stated in line 74. Doing so has not materially changed the resulting correlations, see Figs. 2 and 4.

Line 71: Is it appropriate to use CMIP models. Are they suited for looking at large-scale circulation relationships? Why CMIP5 not CMIP6?

Thank you for this question. Yes, CMIP models are commonly used to study large-scale atmospheric circulation (e.g. Davis and Birner, 2017; Grise and Davis, 2020; Menzel et al., 2019). In many ways, they are more suitable than reanalysis products (Davis and Davis, 2018). Here, we use CMIP5 as the study began before CMIP6 was available.

Line 84/Equation 1: What do the deltas $\delta$ symbolise? Delta is explained later in the text, but should be covered here.

We now describe how $\delta_y$ sets the meridional temperature gradient and $\delta_z$ sets the static stability in the equilibrium temperature profile, see line 97.

Lines 138-140 As mentioned in earlier comment, I suggest you move this paragraph describing season used in this study to be in the methods and abstract.

As you have suggested, we now mention our analysis is limited to the NH DJF at the beginning of the Methods section, see line 66. This is also included in the abstract, line 4.

Line 149, and elsewhere where relevant: There are strong seasonal differences in the HC, STJ, EDJ locations and strengths, how do these differences impact your results for the southern hemisphere vs the northern hemisphere?

Typically, the STJ is weaker in the summer season. Due to this, it is more common for the EDJ to mask the STJ, resulting in moderate and statistically significant negative correlations. I've added this note in line 171, see also discussion in 165 and 251. Likewise, the HC is also weaker in the summer season. For these reasons, our study is focused on the NH winter.

Line 151 and 153: its not it's

See these edits at lines 167 and 168.

Line 185: Are you planning on finding the physical mechanisms responsible for the disconnection? I do not think you come back to this point. I think you can exclude a couple of mechanisms (moist and radiative processes), but what might explain the physical mechanisms?

We believe that the disconnect reflects differing sensitivity of $\phi$HC and $\phi$STJ to the midlatitude eddies. As shown in Fig. 4, a strong coupling between $\phi$HC and $\phi$uv is a robust result regardless of model complexity. In contrast, when an idealized configuration is able to sufficiently simulate a "thermally-driven" STJ, $\phi$STJ becomes less sensitive to the midlatitude eddies, as shown by lower correlations with $\phi$uv, and subsequently becomes decoupled from $\phi$HC. See lines 299-308 in Concluding Remarks for this discussion.

Line 195: 'does not' not 'down not'

See this edit in line 217.

Line 236: I think it would be good to move (or repeat) the physical interpretation of $\delta_z$ in the methods.

Thanks for this suggestion, you will find the physical interpretation both in line 97 and line 104.

Lines 227 – 229 While the correlations do contrast with MB16, they are also quite different to the reanalysis, and I am not convinced being within the range of the CMIP models makes the correlation accurate. Do the statistically significant, if weak, negative correlations between

the STJ and HC, and STJ and uv suggest the STJ is more eddy driven in this model? What are the implications for this moderate, negative correlation?

We interpret the statistically significant negative correlations to reflect instances of the EDJ masking the STJ, as is seen in the coupled model output (i.e. CMIP5 and S-RIP). This is discussed in line 165 but we have reminded the reader of this discussion in line 251.

Line 261: This is the first time that $\delta_z = 30$ has been described as having an improved basic state, and you may wish to state this earlier in the text to make it clearer. How realistic is this $\delta_z$ value? Are there implications for having a much stronger zonal wind in the higher latitudes than the other $\delta_z$ values? The static stability is still much stronger in the tropics than in reanalysis of WR18 or WR18z; is this an issue?

Thanks for these questions. We now state that modifying $\delta_z$ improves the basic state in the Methods section, see line 104. Setting $\delta_z$ to larger values (i.e. $\delta_z = 30$) does simulate more realistic static stabilities and so we do not consider it an issue that the static stability in MB16 ($\delta_z = 30$) is still stronger compared to the WR18 configurations and S-RIP.

Figure 2: The correlations are the model-mean for CMIP5, are they the mean across the 3 reanalyses produces in S-RIP? Please update the caption accordingly. Do you get very different correlations if you look at the individual reanalyses, or individual models?

Yes, the S-RIP correlations in Fig. 2 are the mean across products or "models" as we have stated in the caption. We have edited the caption to include "product-mean" to alleviate any doubt. The standard deviation of correlations across the reanalysis products and coupled climate models is shown in the parentheses under the correlations themselves. This provides a measure of consistency between products and models. You may notice those standard deviations are low demonstrating robustness. This question is also answered by Fig. 4 which shows the correlations for each individual reanalysis product rather than a product-mean.

Figure 3: It is hard to pick the temperature contours from the colour bar, making it difficult to visually compare to the model equilibrium temperatures. Could a more distinct colour bar be used (e.g. with more colours than shades of red)?

We kept the colorbar but changed it's bounds so it is easier to discern the values for the shades, see Fig. 3.

Figure 4. The correlation between Hadley Cell and uv latitudes go off the edge, I suggest you widen slightly. Do you really expect a 100% correlation at $\delta_z = 10$?. I wonder if it's helpful to reverse the order of the metrics such that CMIP5 (or preferably reanalysis) is on top and $\delta_z = 10$ is on the bottom. I intuitively assumed the highest $\delta_z$ was on the top and initially thought increasing static stability made the correlations stronger, the opposite to the real result but an easy mistake to make with a quick glance at the plot. Reversing the order has the added benefit of making the reference value (reanalysis) easier to pick.

Thanks for these suggestions. We have widened the axes so that the correlations no longer are cut off and reversed the order so that the coupled model output (i.e. CMIP5 and S-RIP)

[Figure]

Figure R4: As in Fig. 05 (and Fig. R3) for SH DJF.

are on top, see Fig. 4.

Figure 5: Do you find differences in profiles for the southern hemisphere? It might be nice to add to the supplementary material as you show the southern hemisphere data in the earlier plots. To avoid confusion, I also suggest you add 'vertical' to the dotted and solid line description to clarify you are talking about the STJ and HC latitudes, not the stability or wind profiles. What does the CMIP5 profile look like?

The SH DJF basic state profiles are included in Fig. R4 and as Fig. S2 of the Supporting Document. Typically, the subtropical jet is much weaker in the summer season. One can see that by the moderate $u_{adj}$ values compared to the winter season. Even so, the conclusions found in NH DJF hold for SH DJF; $u_{adj}$ follows the $\partial\theta/\partial y$ profiles, the MB16 configurations simulate $\partial\theta/\partial y$ values of opposite sign as the WR18 configurations and S-RIP, and there are not strong differences between configurations in $\psi_{500}$ and $uv$. Although not shown, the CMIP5 output presents similar profiles as the S-RIP output. See Fig. 5's edited caption to include "vertical."

**References**

Davis, N. and Birner, T.: On the discrepancies in tropical belt expansion between reanalyses and climate models and among tropical belt width metrics, Journal of Climate, 30, 1211–1231, 2017.

Davis, N. and Davis, S. M.: Reconciling Hadley cell expansion trend estimates in reanalyses, Geophysical Research Letters, 45, 11–439, 2018.

Grise, K. M. and Davis, S. M.: Hadley cell expansion in CMIP6 models, Atmospheric Chemistry and Physics, 20, 5249–5268, 2020.

Held, I. M. and Hou, A. Y.: Nonlinear axially symmetric circulations in a nearly inviscid atmosphere, Journal of the Atmospheric Sciences, 37, 515–533, 1980.

Korty, R. L. and Schneider, T.: Extent of Hadley circulations in dry atmospheres, Geophysical Research Letters, 35, 2008.

Maher, P., Kelleher, M. E., Sansom, P. G., and Methven, J.: Is the subtropical jet shifting poleward?, Climate Dynamics, 54, 1741–1759, 2020.

Menzel, M. E., Waugh, D., and Grise, K.: Disconnect between Hadley cell and subtropical jet variability and response to increased CO2, Geophysical Research Letters, 46, 7045–7053, 2019.

Seidel, D. J., Fu, Q., Randel, W. J., and Reichler, T. J.: Widening of the tropical belt in a changing climate, Nature geoscience, 1, 21, 2008.

Shaw, T. A.: Mechanisms of future predicted changes in the zonal mean mid-latitude circulation, Current Climate Change Reports, 5, 345–357, 2019.

Solomon, A., Polvani, L., Waugh, D., and Davis, S.: Contrasting upper and lower atmospheric metrics of tropical expansion in the Southern Hemisphere, Geophysical Research Letters, 43, 2016.

Waugh, D. W., Grise, K. M., Seviour, W. J., Davis, S. M., Davis, N., Adam, O., Son, S.-W., Simpson, I. R., Staten, P. W., Maycock, A. C., et al.: Revisiting the relationship among metrics of tropical expansion, Journal of Climate, 31, 7565–7581, 2018.

---

## Referee Report (RR1)

Review of

**Replicating the Hadley Cell edge and subtropical jet position disconnect in idealized atmospheric models**

Molly Menzel et al.

**General**

This is my second review of this work, where the authors employ a model hierarchy approach to examine the disconnect between interannual variations of the position of the subtropical jet and the edge of the Hadley cell — showing that this disconnect can be reproduced in an idealized dry GCM. The authors have appropriately addressed my concerns and I recommend the work for publication. A few minor comments and suggestions are listed below.

**Comments by line number in the tracked changes version**

7    Indeed, the usage of 'accuracy' in the context of an idealized model is problematic. But 'sufficient' here is obscure. Sufficient in what aspect? Perhaps 'sufficiently **realistic** climatological basic state'?

8    not clear what are the 'features' being referenced. The metrics?

19   Add space: $CO_2$ (Davis…

60   same as in line 7

67   analyses

101  treating $\varepsilon$ and $\chi$ as two separate parameters in $\varepsilon\chi \sin\phi$ when in effect $\varepsilon\chi$ is a constant seems redundant, as both $\varepsilon$ and $\chi$ have the same influence on $T_{eq}$.

---

## Referee Report (RR2)

Replicating the Hadley Cell edge and subtropical jet latitude disconnect in idealized atmospheric models
Molly Menzel, Darryn Waugh, Zheng Wu, Thomas Reichler

General comments:
Menzel et al. use reanalysis and several models of varying complexity to explore the disconnect between the Hadley Cell edge and the location of the subtropical jet in DJF. They find that the disconnection can be simulated without moist or radiative processes or a zonally asymmetric state. Based on correlations, they argue that the disconnection is due to different sensitivities to midlatitude eddies, with the Hadley Cell edge being very sensitive to mid-latitude eddies, and the subtropical jet location less so. This is a really interesting study and helps address a gap in our dynamical understanding of the global circulation.

My concerns from the previous draft have been addressed, thank you. I have made a few, final, minor comments below. Otherwise, I think this manuscript is ready for publication and will make a useful contribution to the literature.

Minor Comments:

Line 28: this sentence is a bit confusing to follow. Maybe a comma after 'is limited' would make it easier to read.

Line 47 and elsewhere: its not it's

Line 219/Figure 4: It would be good to remind the reader which is the default experiment in MB16 (delta = 10) to make it easier to compare results between figure 3 and 4. Noting that the correlation is marked in red is helpful, but insufficient as all MB16 experiments are similar red or orange shades, so it is not immediately clear which one(s) you are referring to.

Fig 5. The caption should include that these values are zonal averages.

---

## Author Response (AR2)

**Response to Reviewers**

Authors' responses to reviewers' comments are distinguished by green text.

**Reviewer 1**

General

This is my second review of this work, where the authors employ a model hierarchy approach to examine the disconnect between interannual variations of the position of the subtropical jet and the edge of the Hadley cell — showing that this disconnect can be reproduced in an idealized dry GCM. The authors have appropriately addressed my concerns and I recommend the work for publication. A few minor comments and suggestions are listed below.

We appreciate that you've taken the time to review this paper a second time. Thank you for recommending the paper for publication.

Comments by line number in the tracked changes version

7 Indeed, the usage of 'accuracy' in the context of an idealized model is problematic. But 'sufficient' here is obscure. Sufficient in what aspect? Perhaps 'sufficiently realistic climatological basic state'?

Thanks for your note, we have implemented your suggestion in line 7.

8 not clear what are the 'features' being referenced. The metrics?

Agreed, we have edited the sentence in line 8 to be "... the robust disconnect between $\phi$STJ and $\phi$HC across the model hierarchy reveals their differing sensitivities to midlatitude eddy momentum fluxes...".

19 Add space: CO2 (Davis...

See this edit in line 19.

60 same as in line 7

See the inclusion of "sufficiently realistic" in line 59.

67 analyses

See this edit in line 66.

101 treating $\varepsilon$ and $\chi$ as two separate parameters in $\varepsilon\chi\sin\phi$ when in effect $\varepsilon\chi$ is a constant seems redundant, as both $\varepsilon$ and $\chi$ have the same influence on $T_{eq}$.

Yes, we agree that for our purpose, it is redundant. However, we choose to remain consistent with prior work by keeping them as independent parameters, see McGraw and Barnes (2016) and Chen and Plumb (2014).

**Reviewer 2**

General comments:

Menzel et al. use reanalysis and several models of varying complexity to explore the disconnect between the Hadley Cell edge and the location of the subtropical jet in DJF. They find that the disconnection can be simulated without moist or radiative processes or a zonally asymmetric state. Based on correlations, they argue that the disconnection is due to different sensitivities to midlatitude eddies, with the Hadley Cell edge being very sensitive to mid-latitude eddies, and the subtropical jet location less so. This is a really interesting study and helps address a gap in our dynamical understanding of the global circulation.

My concerns from the previous draft have been addressed, thank you. I have made a few, final, minor comments below. Otherwise, I think this manuscript is ready for publication and will make a useful contribution to the literature.

Thank you for supporting publication of our manuscript.

Minor Comments:

Line 28: this sentence is a bit confusing to follow. Maybe a comma after 'is limited' would make it easier to read.

We broke up the sentence with a semicolon so that it is easier to understand, see line 28.

Line 47 and elsewhere: its not it's

Thanks for catching this typo! We have combed through the manuscript to correct any additional mistakes (lines 47, 94, 100, 233, 259).

Line 219/Figure 4: It would be good to remind the reader which is the default experiment in MB16 (delta = 10) to make it easier to compare results between figure 3 and 4. Noting that the correlation is marked in red is helpful, but insufficient as all MB16 experiments are similar red or orange shades, so it is not immediately clear which one(s) you are referring to.

Thanks for suggesting this clarification. We specify that we are referring to the default parameter of $\delta_z = 10$, marked by the dark red in Fig. 4, see line 219.

Fig 5. The caption should include that these values are zonal averages.

We have ensured all figure captions specify "zonal-mean" where appropriate.

**References**

Chen, G. and Plumb, A.: Effective isentropic diffusivity of tropospheric transport, Journal of the Atmospheric Sciences, 71, 3499–3520, 2014.

McGraw, M. C. and Barnes, E. A.: Seasonal sensitivity of the eddy-driven jet to tropospheric heating in an idealized AGCM, Journal of Climate, 29, 5223–5240, 2016.